# The mediating role of green innovation and green culture in the relationship between green human resource management and environmental performance

Liuyue Fang[1]*, Shengxu Shi[2], Jingzu Gao[1], Xiayun Li[1]

**1** UCSI University, Kuala Lumpur, Malaysia, **2** College of Public Management, Fujian Agriculture and Forestry University, Fuzhou, Fujian, China

* fly0612@sina.com

**Data Availability Statement:** All relevant data are within the manuscript.

**Funding:** The author(s) received no specific funding for this work.

## Abstract

There is evidence in the literature that green HRM practices improve environmental profitability. On the other hand, existing research has failed to explain how GHRM can support the development of a green culture and green innovation influence the firm's environmental performance and long-term growth. This study investigates the relationship between GHRM, green culture, green innovation, and a firm's environmental performance. In addition, the study examines the mediating role of green culture and green innovation in the relationship between GHRM and environmental performance. This research conducts a large-scale study of 290 employees from Manufacturing firms in Malaysia. The research results provide managers with a better knowledge of how GHRM helps develop sustainable culture and green innovation and how these elements contribute to the improvement of environmental performance inside the organization. This study also makes a significant contribution in terms of novelty and research relevance by demonstrating that green culture and green innovation positively mediate the relationship between GHRM and environmental performance in a sustainable manner. Managers will understand the GHRM required to develop an ecologically conscious culture and promote green innovation among environmentally conscious employees. Finally, we highlighted the importance of this study for top management in the sense of mediating the role of green culture and green innovation and the consequences for future generations of responsible managers who will acquire this knowledge.

## Introduction

Environmental degradation and climate change have increased stakeholders' anxiety about the prevalence of environmental challenges, resulting in higher levels of global warming. At various levels, efforts are made to address the conflict between environmental concerns and long-term growth strategically. Since the turn of the century, environmental rehabilitation has become more significant. Over the last several decades, researchers have paid more attention to an organization's environmental performance. The Malaysian government shows a strong

**Competing interests:** The authors have declared
that no competing interests exist.

focus on environmental performance as part of its policies to protect natural resources and the
environment and accomplish sustainable development objectives [1]. Environmental perfor-
mance is critical in protecting the natural environment from negative effects such as pollution,
environmental emissions, and wastes and maintaining organizational performance [2]. As
addressing environmental challenges fosters organizational culture and innovation develop-
ment, firms' environmental performance has become a viable source of competitive advantage
[3]. However, according to a recent study, there has been little research on environmental per-
formance in the Malaysian context [4].

In light of this reality, green human resource management has been more popular in the
last decade as a proactive approach that organizations can adopt to improve their environmen-
tal performance [5]. Green HRM is a collection of human resource management techniques
that take the organization's environmental manifesto into account [6]. Developing an employ-
ee's green capabilities entails assimilation conclusive environmental thinking via human
resource processes such as leadership development, training, recruiting, and selection [7].
Employees who have been recruited and trained continue to be motivated by performance
measurement and remuneration systems that emphasize chances for environmental perfor-
mance enhancement [8]. Numerous researchers have examined the link among green human
resource management (GHRM) practices and the environmental performance (EP) of organi-
zations [9–11]. These researchers find that green HRM practices positively affect a firm's envi-
ronmental performance by reducing waste and organizational efficiency [8]. In general, green
HRM practices can help employees adopt more environmentally friendly behaviours, volun-
tarily improving organizational performance [2, 12].

Nevertheless, admitting the relationship among green human resource management prac-
tices and environmental performance is widely known, we believe that any research on how
environmentally aware employees execute green initiatives without considering green culture
is incomplete. Furthermore, a recent study has shown a insufficiency of study on the link
between green culture and an organization's environmental performance [13–15]. [13] argued
that the mediating role of green culture in the relationship between green human resource
management and environmental performance is currently unclear. In line with this [13], argue
that the interaction of green human resource management and green culture is a critical area
of study for academic scholars. This study will address these gaps by addressing the following
research objective: How can green human resource management practices and a green culture
impact an organization's environmental performance?

Recent studies have explored the consequence of green HRM on a firm's environmental per-
formance and identified a positive relationship [16]. Even though scholars have proved the
effect of green HRM on employee and organizational outcomes, there is still a need to investi-
gate how and when green human resource management affects green organizational innovation
[17]. Although past research has explored the positive effect of green human resource manage-
ment practices on green innovation [18]. According to [19], green HRM practices that address
systems and methods for influencing employees in a controlled way on a larger scale could be
the greatest indicators of increased green innovation and environmental performance. How-
ever, little experimental consideration has been paid to the link among green HRM and green
innovation. We address this research gap in the literature by examining how green innovation
affects the link between green HRM and environmental performance by using it as a mediator.

The purpose of this study is to formulate a comprehensive research model that demon-
strates how green HRM improves a firm's environmental performance via intervening role of
green innovation and green culture. The remainder of this section will go as follows. First of
all, we develop a study framework and propose seven hypotheses. Second, we explain the
methodology and measurement of the variable. The hypotheses are then tested using structural

equation modelling and the bootstrapping test. Finally, we examine the theoretical and practical contributions and the limitations of the research and suggest future research directions.

## Literature review and hypothesis development

### Green HRM and environmental performance

Various studies have been conducted to determine how pro-environmental HRM practices increase a firm's environmental performance [20, 21]. Environmental performance is explained systematically as an organization's commitment to safeguard the environment and exhibit quantifiable operational metrics within the required standards of environmental stewardship [22]. [23] present a systematic measure of environmental performance, encompassing recycling performance, incident devaluation, unceasing enhancement, waste reduction, stakeholder awareness, independent examination, resource consumption devaluation, and cost accumulation. Human resource managers are necessary to reach these environmental performance goals by recruiting, educating, evaluating, and rewarding an environmentally conscientious staff [24]. The environmental credentials of their organization are often aggressively promoted by human resource managers in order to attract job applicants looking for organizations that share their values and beliefs [25]. Recently graduated are entering the workforce, and many are searching for positions in environmentally friendly organizations [26]. It is becoming more common for human resources managers to include environmental awareness requirements in job descriptions and interview procedures in order for the firm's environmental goals to be met by future workers [27]. It is the responsibility of human resources managers to train both operational and managerial staff. The selection and promotion of environmentally conscious leaders is a critical function of human resources [28]. Environmental organization leaders typically require transformational leadership and transactional management skills [29]. Human resource managers must look for and retain leaders who can efficiently shift between critical and operative decision-making exercise [30]. In order to improve the firm's environmental performance, leadership will advocate for ecologically-focused activities [31].

Employee performance can be evaluated by looking at how well the organization is doing regarding environmental objectives. Managers have the ability to create and use environmental performance indicators and metrics throughout the whole organization [32]. In the course of performance evaluations, managers must discuss with their staff whether they have met their environmental aims and recommendations for waste diminution and performance improvement that they may have [33]. While organizational employees are usually motivated by minimizing environmental impact, their attitude can be affected further via compensation and incentive schemes [34]. Numerous researchers studies have established a relationship between higher management benefit and a firm's environmental performance [35–37]. [37] concluded that senior executives' cash compensation clearly correlated with organizational environmental performance in a survey of 698 firms.

Similarly, [36] showed that the involvement of a senior environmental manager positively moderates the relationship with corporate environmental performance. A study of the literature on green HRM practices reveals that environmental performance is positively determined by recruiting, retention, and appraisal. As a result, the following is our hypothesis:

*H1*: *Green HRM practices positively impact organizational environmental performance.*

### Green GHRM and green innovation

Green innovation is defined as innovation that has the potential to minimize environmental consequences while still reaching a firm's environmental objectives and generating

environmental benefits [38]. Prior research has shown that human resource management can help organizations improve their workers' knowledge, skills, and capabilities, hence promoting product and process innovation [39]. On this basis, we argue that green human resource management positively affects green innovation. To begin, green recruiting increases the firm's environmental management effectiveness since employing more environmentally conscious staff results in their participation in higher eco-friendly exercise [40]. Employees with a high degree of environmental aptitude and awareness can generate more and more unique and beneficial objective for environmental management, increasing the firm's green innovation. Thus, organizations should select employees committed to environmental responsibility to foster and maintain green innovation [41].

Furthermore, internal firm training and participation procedures can allow employees to acquire the information and capabilities necessary to boost their inventive innovation [42]. Specifically, when an organization depends on green training and engagement, it can stimulate the development of new ideas for green innovation, including product or process innovation [43]. Employees who get green training are brilliant and talented better able to detect environmental disputes and are likely to engege in relevant activities that recommend green innovation. Additionally, green involvement can foster green employee behaviour and give an opportunity for organizational employees to avail their both green knowledge and abilities [44], supporting green innovation.

Moreover, green performance management and incentives policies can assist employers in aligning employee behaviour with the corporate environmental objectives [45]. While green innovation is the component of environmental management that precisely tackles environmental concerns, green performance management is an excellent technique to increase employee environmental engagement and hence their readiness to commit in eco-friendly innovation [46]. Additionally, invigorate environmental attempt and innovative ideas for green goods and processes can boost foster an innovative culture under organizational walls [17]. Within each organizational, top managers should motivate employees to be innovative in developing green products and processes without fear of failure [47]. Finally, the HRM previous studies reveal that HRM systems can have a positive effect on product and as well as on the process innovation [47, 48], which means that combining HR practices can have a greater impact on innovation than using individual practises alone [40]. As a result, we analyze the three components of green human resource management practises as their whole, as they all subsidize to green innovation. According to the logic above raised, GHRM can help employees develop their capability, motivation, and opportunities, increasing their resources and capabilities of the green product and as well as the green process innovation. As a result, we suggest the following hypothesis:

*H2*: *Green HRM positively affects green innovation.*

## Green human resource management and green culture

Organizational culture can be considered green if its employees can minimize negative environmental effects while maximizing positive environmental benefits by going above and beyond profit-seeking purposes [49]. There is increasing demand to develop organizational policies that promote green behaviour. As a result, the human resource function incorporates pro-environmentalism and ecological concepts into practically all activities and procedures [50]. This finally results in employees adopting a more environmentally conscious attitude [50] and the reduction of environmental waste and development of a green culture [51]. Human resource management (HRM) plays a vital role in supporting the firm's green culture

by influencing employees' attitudes, behaviours, and values via recruiting, training, performance management, and incentive [52].

[7] recent study found that organizations must establish human resource practices that contribute to increasing employee engagement and behaviour that supports organizational transformation for sustainable growth. Indeed, [22] found that trained and rewarded workers for encouraging pro-environmental activities conclusively contributed to creating and promoting a green organizational culture. [53] published a comprehensive study on the enablers of effective green manufacturing adoption in Irish enterprises. The research identified both interior and exterior facilitator of green manufacturing methodly. Internal facilitator included organizational culture. While the study recognized organizational culture as an enabler of green manufacturing, scholars failed to advance the study by examining the enablers of green culture and their mediating part in managing environmental performance for Malaysian manufacturing firms.

According to [54], four indicators contributed to an unexpected transition in American culture. Employee empowerment, message believability, leadership focus, and peer involvement are all examples of these. Even though the four indicators promoted a transition toward quality management [54] (Muisyo et al., 2021), we suggest that they have the potential to allow an organization's green culture. This study hypothesis is justified by [55], who suggested that environmental management and quality management control are inextricably linked systems that, when integrated, can boost an organization's productivity. This hypothesis is also confirmed by [2] which demonstrated that the above-mentioned aspects contributed to a firm's green performance. However, [8, 54] studied green culture as an aggregate variable rather than disaggregating it into its constituent components and examining how each aspect contributes to environmental performance. This research aims to examine the mediating role of green culture in the relationship between green human resource management and environmental performance.

Pro-environmental leadership emphasizes the need of effecting the environment a primary concern for organizational superior authorities who practice pro-environmental efforts in their everyday activities and assess the employees based on their environmental performance [56]. Human resource managers are accountable for hiring environmentally conscientious personnel and advancing them to leadership positions [57]. Additionally, human resources can incentivize executives to undertake environmental initiatives by tying compensation to environmental performance improvement [41, 58]. A pro-environmental encouragement scheme manage at organizational higher leaders then cascades from top to bottom the hierarchy as leaders establish environmental goals for each section and its subordinates [59]. [54] highlighted that message credibility is given by reputable sources that are harmonious, fast and effortless to understand, and personally engaging to employees. Human resource managers are uniquely positioned to design pro-environmental messaging that respond to employees' concerns about decreasing inefficient and ecologically damaging actions in their everyday duties [60]. The human resource department may transmit employee pro-environmental ideas via training sessions and performance review meetings [27].

Peer involvement refers to employee engagement and correlative support for environmental efforts [61]. Through training and incentive systems, human resources can foster a culture of peer involvement in environmental initiatives [62]. Human resources can collaborate with firm's higher level managers to establish primary performance measures for organizations responsible for delivering pro-environmental activities. The key performance indicators can be related to waste depletion efforts, increased reprocesing, and resource consumption reductions [8]. By attaching financial incentives to the achievement of key performance indicators, human resource managers can motivate workers to collaborate with peers on environmental projects [62]. Organizational employee empowerment indicate to the degree to which an

employee has the ability to make appropriate decisions in situations and circumstances that go beyond basic norms [63]. [64] argue that environmental empowerment increases workers' environmental consciousness. Managers and workers gain empowerment due to human resource-focused efforts, such as training and evaluation [65]. Employees who work under empowered managers who guide by example are more credible to embrace environmental change and take proactive steps to decrease detrimental organizational procedures [66]. Employees who go above and beyond the duty standard might get extra money during performance assessments. Additionally, human resources can motivate workers to solve environmental challenges via methods for instance green teams. Organizational team members play critical parts in recognizing and addressing concerns throughout collaborative efforts [66].

We can perceive already in the previous literature how green HRM strategies contribute to the development of a green culture. The human resource management department recruits environmentally aware personnel through training, leadership, and rewards and develops pro-environmental ideas and perspectives. These attitudes and beliefs reveal themselves in an employee's everyday work as pro-environmental behaviours. As workers engage and collaborate to address environmental concerns, these behaviours become routines, and an organization develops a pro-environmental culture. Based on this knowledge, we propose that green HRM practices have a beneficial consequence on the improvement of leadership significance, message credibility, peer involvement, and employee empowerment; in other words, on the development of the green culture. This arguments leads us to the following hypothesis:

*H3*: *Green human resource management practices positively relate to green culture.*

## Green innovation and environmental performance

The environmental performance focuses on the organizational actions that go beyond primary compliance with basic rules and regulations to meet and exceed social expectations about the natural environment [67–69]. It involves the environmental impacts of organizational operations, products, and resource use in the most compliant way possible with applicable environmental laws [70]. According to previous research, environmental performance is based on the standard of environmentally friendly goods, the development of green processes and products, and the integration of ecological, eco-friendly practices into corporate operations and product development [19, 71–73].

Green innovation is linked to a firm's environmental management strategy, which improves environmental performance [19, 74]. Additionally, green product and process innovation mitigate an organization's negative environmental effect and improve its economic and social performance via waste and cost reduction [75]. According to previous research, green innovation should not be perceive as a firm's reactive feedback to stakeholder demands but rather as motivated organizational intents and operations to improve environmental performance in order to obtain a competitive edge [68, 76–78]. Based on prior research, we expect that green process and product innovation are significant organizational assets that firms use to improve their environmental performance and gain the trust of key stakeholders [77]. As a result, we hypothesize that:

*H4*: *Green innovation positively influences environmental performance.*

## Green culture and environmental performance

Green culture is a contemporary environmental philosophy based on aesthetics that promote sustainable economic and ecological growth. Several years ago, organizations started

incorporating this technique into their corporate social responsibility efforts. Organizations recognized that this paradigm change would alter market behaviour, resulting in increased sales and profitability [79]. This environmentally aware shift allowed for new ideas, all of which pushed the organization toward sustainability or social consciousness in line with its principles. In other words, using this green strategy would boost organizational culture. We then hypothesize that a green culture can result in an improvement in the organization's environmental performance. We argue that four characteristics of culture, namely leadership emphasis, message credibility, employee empowerment, and peer involvement, can all benefit environmental performance improvement criteria [8].

For instance, a proactive perspective on environmental issues, with a strong reliance on leadership, has been demonstrated to assist employees to recognize environmental concerns and equip them to instrument positive environmental approaches, such as reprocessing and development programmes [80, 81]. According to [82], environmentally aware management team members can take proactive environmental measures by matching environmental and financial objectives [83]. Senior leaders communicate proactive environmental measures to operational staff, which become integrated into their day-to-day activities over time [84]. Thus, making the environment a primary focus as a leader demonstrates in employees' pro-environmental behaviours, enabling them to concentrate on process improvement projects such as eliminating extravagant activities from the manufacturing process [85]. As a result of decreasing and reshaping raw materials, reprocessing performance is improved, resource utilization is reduced, and expenses are reduced [8]. Additionally, credible pro-environmental messaging from top management motivates environmentally concerned personnel to behave responsibly [86]. More precisely, messaging that aligns with an staff member aim to mitigate environmental damage might influence how employees convey pro-environmental performance to collaborator [87–89]. Improving stakeholder views of an organization's environmental performance might help the organization score higher in sustainability indexes and attract further investment [90].

Peer involvement can help develop cooperation activities around the organization's environmental goals [91]. Environmentally aware collaboration is supposed to significantly decrease waste and improve an organization's environmental performance [92, 93]. For example, [94] indicate that organizations can only achieve the proactive stage of environmental management when teams embrace pro-environmental thinking. Furthermore, [95] suggest that peer involvement and environmentally aware cooperation are critical components of green integration. Teams can concentrate on continual enhancement projects to lower hazardous emissions and excessive waste throughout the manufacturing process or on programmes targeted at minimizing the frequency of adverse environmental occurrences within an operation [96, 97]. When employees are empowered to make their own choices, they can recognize and promptly remedy damaging actions inside a business. For instance, employees can be empowered to recognize procedure that use uncontrolled raw materials and build proactive recycling programmes to lower inclusive consumption assessment [98].

Additionally, employees can be empowered to conduct examines of their individuals and their peers' procedures to foster a culture of ongoing pro-environmental improvement. Indeed, [73] have revealed that organizational employee empowerment increases employees' awareness of environmental issues and can significantly and conclusively affect the organization's environmental performance. Based on this argument, we hypothesize that green culture is the key to environmental performance. In addition, This leads us to hypothesize the following:

*H5*: *Green culture can positively influence an organization's environmental performance.*

## Green innovation mediates the relationship between green HRM and environmental performance

Green HRM improves employees' environmental consciousness [99], green innovativeness [100], and green organization performance [101]. Previous research indicates that green HRM affects green innovation [102] and green firm performance [103]. Still, these areas of examination remain largely unexplored and require additional empirical research, particularly as organizations face increased pressure from key stakeholders to adopt eco-friendly management practices. Additionally, [33] propose recruiting prospective workers based on their environmental ideas, attitudes, and expertise. By leveraging potential employees' environmental ideas, values, and expertise, businesses can ensure that new hires realize and comprehend the organization's environmental principles and values [2]. Similarly, green training and development [104], performance management and appraisal [19], and green rewards and compensation [11] stand out as critical HRM practises that provide to superior environmental performance.

Earlier research indicates that human resource management systems affect innovation [105, 106]. Human resource management systems, we argue, have an effect on administrative, process, and product innovation [107]. Additionally, human resource management practices foster employee commitment more than conformity with organizational rules and systems [108]. Additionally, [107] argue that commitment and collaborative effort human resource management practises having a differential effect on firm innovation, with the former increasing interior innovative abilitioes and the latter encouring innovation through the institution and nurturing of social networks with external sources. On the other hand, green innovation is a critical asset for environmental performance [109], which firms employ to accomplish their environmental management objectives. Green product and process innovation significantly reduces the organization's negative environmental impact, if there are any, and improves firm performance across all dimensions, including financial, social, and environmental performance, through significant waste and cost reduction that saves money, time, and resources [110]. As a result, we hypothesize that green human resource management will indirectly affect firm environmental performance through the mediating role of green innovation. As a result, we propose the following hypothesis:

*H6*: *green innovation mediate the relationship between green HRM and environmental performance.*

## Green culture mediates the relationship between green HRM and environmental performance

After developing green HRM, firms are expected to communicate the organization's environmental preservation philosophy and regard to their staff. Once employees recognize the importance of green human resource management, green culture becomes a major mechanism for improving environmental performance [19, 90, 111]. Indeed, the strength of green culture is contingent upon employees agreeing on their assessment of the situation in which they find themselves [62, 90]. Thus, a solid green culture can emerge when corporate personnel share environmental values, attitudes, and practices [112]. This results in a mutual understanding of the environment. The teams then go beyond profit motives and work together to improve environmental performance, arising in a stable organizational setting that impacts the organization's overall environmental performance [70].

This is also accomplished through greening hiring, incentives, performance development, and training [90]. For example, if a firm has a green green culture, upper management would urge employees to obtain green awareness and engage in a discussion about environmental

problems. They work together to address environmental issues and share a sense of responsibility to preserve the environment, therefore fostering green culture. According to [89], an employee-centered green culture tends to foster green behaviours that lead to more exceptional environmental performance as well as enhanced social and psychological happiness among employees. As a result, according to [11], green culture is a critical link between green human resource management and environmental performance. For example, if pro-environmental incentive and promotion systems were in place, employees would most likely embrace green values, attitudes, and behaviours in order to gain from their efforts to advance in their careers [113]. Similarly, effective pro-environmental leadership, message credibility, peer participation, and employee empowerment are more likely to foster green culture, which drives environmental performance [8]. It is reality that HR departments in businesses play an important role in propagating these ideals via training and performance evaluation sessions [54].

Moreover, rewarding environmental performance as portion of organizational activities motivates employees to collaborate with their peers to meet or surpass the team's and department's key environmental performance indicators [114, 115]. Additionally, [116] argues that including peers and empowering employees in decision-making raises employees' environmental awareness and encourages them to adopt environmentally friendly activities. Similarly, green human resource management (GHRM) efforts that recognize and reward environmentally conscious team members help foster a green culture [45], which eventually results in green behaviours. Green HRM is intrinsically pro-environmental, creating green behaviours and, eventually, environmental performance. As a result, we hypothesize the following (Fig 1):

*H7*: *Green culture mediates the relationship between green HRM and environmental performance.*

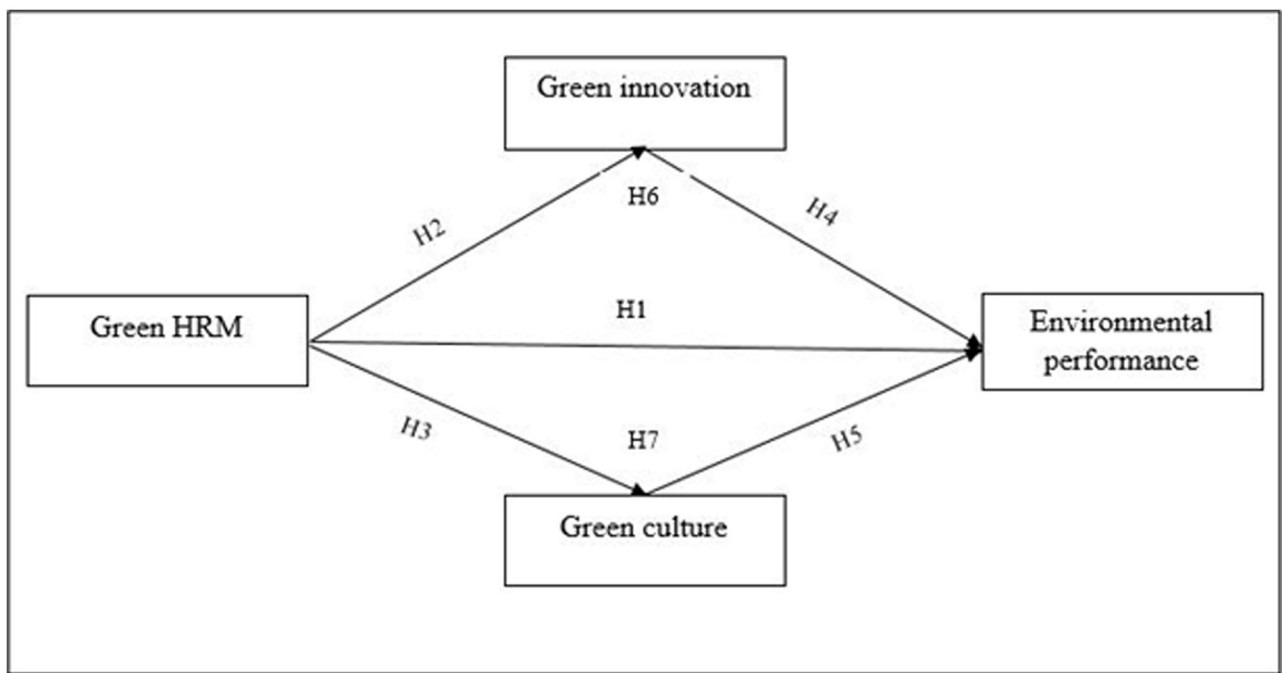

**Fig 1. Study framework.**

## Methodology

### Sample and data collection

Malaysian manufacturing sector employees were included in the study's sample population. Selected sample employees are those employed by firms registered with the Malaysian investment development authority (MIDA) and situated in the country's Penang, Johor and Selangor states. More than 50% of all those who participated in the questionnaire came from Selangor, a Malaysian state with a burgeoning economy and a reputation for high levels of pollution emissions [117]. Because the Malaysian manufacturing industry has a shockingly bad environmental record, we chose it in this study [118]. Due to Malaysia's fast industrialization over the previos several decades, which has resulted in a noticeable increase in pollution, many environmental issues have arisen [119]. The public concern prompted the Malaysian government to implement tough regulations to lower the quantity of inhalation exposure particles and has urged industrial enterprises to limit coal consumption, implement green exercise and remove significant pollution causes [120].

The survey instrument was intended to elicit responses on four major constructs of our study: GHRM practises, GI, GC, and EP. We obtained 290 legitimate replies to 500 questions, representing a 58 percent response rate. Human resource (HR) managers accounted for 54% of responses, while IT employees accounted for 24%. This sample is ideal for our research since it includes both managers and employees from a number of organizational areas. Two hundred ninety organizations were randomly selected to participate in our study. We chose head offices as our primary respondents because our research requires respondents to clearly understand the interconnections between GHRM, GI, GC, and EP. This helped in ensuring that we received high-quality answers. We provided the questionnaires by e-mail in sensitivity to the Malaysian government's Movement Control Order (MCO), which was imposed in light of rising COVID-19 cases in these states.

### Measurements

Every item in the four key constructs of this study was assessed on a five-point Likert scale ranging from 1 (strongly disagree) to 5 (strongly agree). Green human resource management (GHRM) practises include eight items, as per [33]. Standard and validated statistical measures were used to measure the green culture in the survey instrument based on [8] 's study, there were a total of 10 items for green culture, and the measurements were gathered from a variety of research sources. According to earlier studies, such as [23, 121], environmental performance measures include eight items. A 6-item derived from [40, 120] was used to quantify green innovation.

According to [122], a sample size of less than 50 is weak, a sample size of 51–100 is weak, a sample size of 101–200 is sufficient, a sample size of 201–300 is good, a sample size of 301–500 is very good, and a sample size of 500 or more is excellent. A sample size of more than 520 was employed in this research, which is regarded as significant. Only 307 of the 520 surveys that were provided to managers were returned out of 450. In addition, 17 questionnaires were neglected from the analysis because of insufficient data. That's why Table 1 only includes data from 290 questionnaires in the end. Data from 290 organizations was employed to conduct the final study. As a result, the sample size is appropriate. The sample consists of respondents who are diverse in terms of gender, age, education, work experience, and employment position, among other characteristics. Males were 62% of those who answered the survey questions. Among those who responded, 32% were between the ages of 40 and 50. Approximately 46% of those who answered the survey have at least a Master's degree. The majority of respondents

**Table 1. Profile of respondents.**

| Attributes | Option | Frequency | Percentage (%) | Total |
|---|---|---|---|---|
| Gender | Male | 180 | 62 | 290 |
| | Female | 110 | 38 | |
| Age | 20–30 | 39 | 13 | 290 |
| | 3–40 | 77 | 27 | |
| | 40–50 | 92 | 32 | |
| | 50 –above | 82 | 28 | |
| Education | Junior college | 15 | 5 | 290 |
| | Bachelor's degree | 78 | 27 | |
| | Master's degree | 134 | 46 | |
| | PhD | 63 | 22 | |
| Work Experience (years) | 1–5 | 47 | 17 | 290 |
| | 6–10 | 36 | 12 | |
| | 15–20 | 146 | 50 | |
| | 20 or above | 61 | 21 | |
| Position | IT Manager | 75 | 26 | 290 |
| | HR Manager | 157 | 54 | |
| | Finance Manager | 39 | 13 | |
| | Sales Manager | 19 | 7 | |

50%, had between 15 and 20 years of work experience, with just 21% having more than twenty years of work experience overall. Furthermore, when it came to the job position in this study, 54% of respondents identified themselves as human resources managers.

## Results and data analysis

### Measurement model

For the present study, we employed PLS-SEM to identify the research study model since the PLS-SEM approach has been demonstrated to be effective of handling basic and complicated framworks. Similarly, it works with data that does not meet the standards for normalcy and complexity in the analysis [123]. Related to the covariance-based technique CBS-SEM, PLS-SEM is more accurate in assessing and determining variable validity [123]. The measurement model Fig 2 and a structural model Fig 3 were both assessed with the help of PLS-SEM in the present research work. The research found various validity approaches to estimate the measurement model, including convergent as well as discriminant validity techniques [124]. For this study, all of these parameters fulfil the standardized criteria, which have been defined by several academics and are presented in Table 2.

Convergent validity is the degree to see which variables' items measure the same variable [125]. According to [126], convergent validity evaluates if all constructs' items accurately represent their accompanied predictor. Convergent validity was assessed in order to identify two techniques: average variance extracted (AVE), and composite reliability (CR). Additionally, factor loadings, AVE, and CR values should be greater than 0.50, 0.50, and 0.60, respectively [124]. Cronbach's alpha value should be more than 0.60, according to [127]. In Table 2, it can be shown that the AVE has values more than 0.50, and the CR value is greater than 0.60, as mentioned by [124]. Furthermore, Cronbach's alpha value is larger than 0.60, which is higher than the 0.60 indicated by [127].

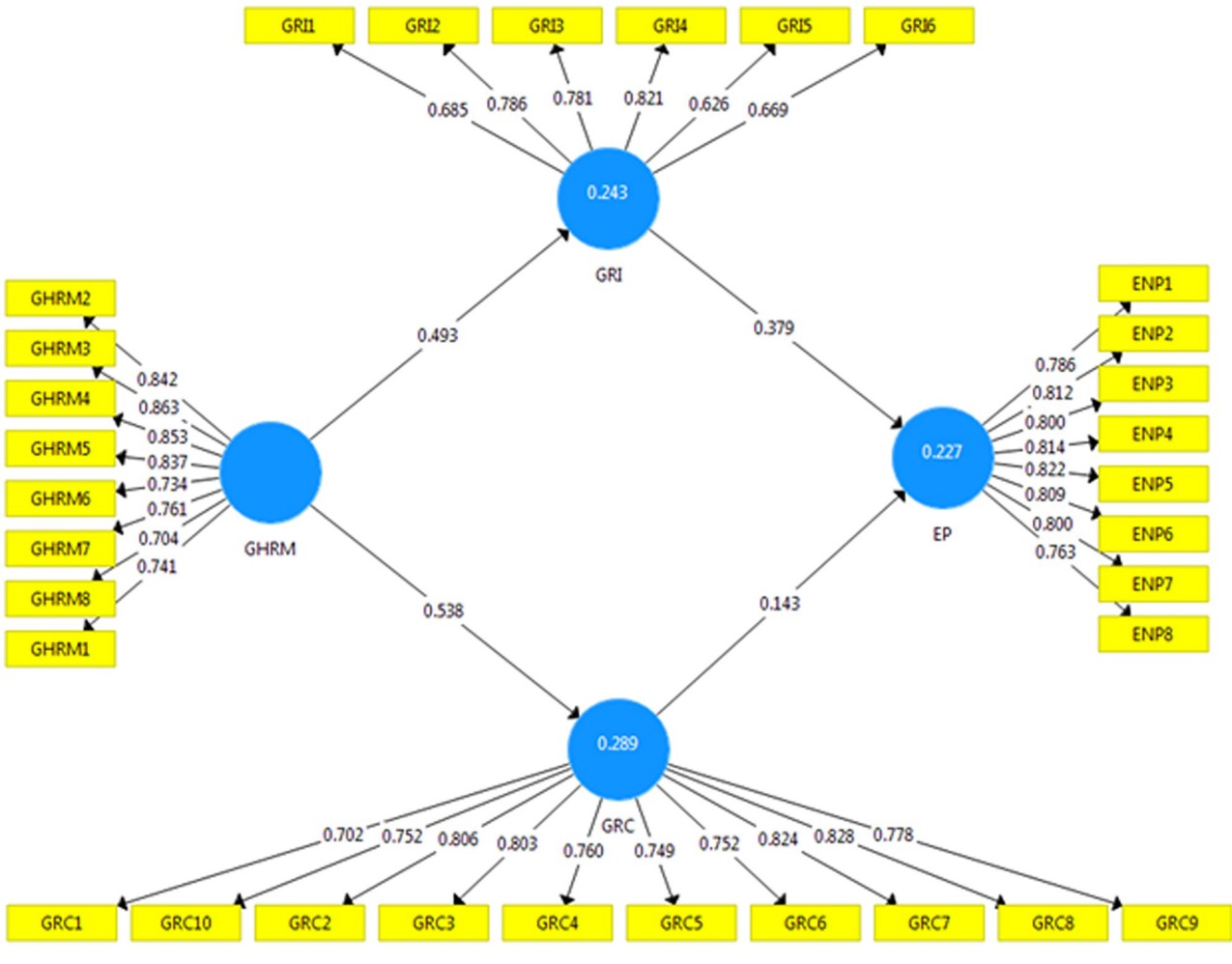

**Fig 2. Measurement model.**

According to statistical definitions, discriminant validity is a circumstance in which a study evaluates two statistically distinct elements [67]. To evaluate discriminant validity, first compute the square roots of AVE and then compare them to the correlations of both of these variables in the theoretical model [128]. Furthermore, the diagonal values of comprized constructs should be higher than the values in the same rows as well as with columns in both rows and columns [129]. As we can see that in Table 3 illustrates, on the other hand, that the present research meets the criteria for discriminant validity. Fornell and Larcker ([129]) proposed that any diagonal upper values more than other related values in the same columns and rows are shown in the previously stated Table 2, which is also included in the following Table 3.

In the past, discriminant validity was assessed using standard metrics developed by Fornell and Larcker in 1981 and based on their findings. Because of deficiencies in standard metrics, several studies proposed a new approach to calculate discriminant validity, such as heterotrait–monotrait (HTMT) discriminant validity estimation [130]. When the difference between loadings is less, the usual measures of discriminant validity are not appropriate techniques to use [130]. The HTMT significance level is 0.90 for conceptually identical constructs, while the HTMT significance value is 0.85 for conceptually dissimilar constructs [130]. According to Table 4, there are no problems with discriminant validity in this research.

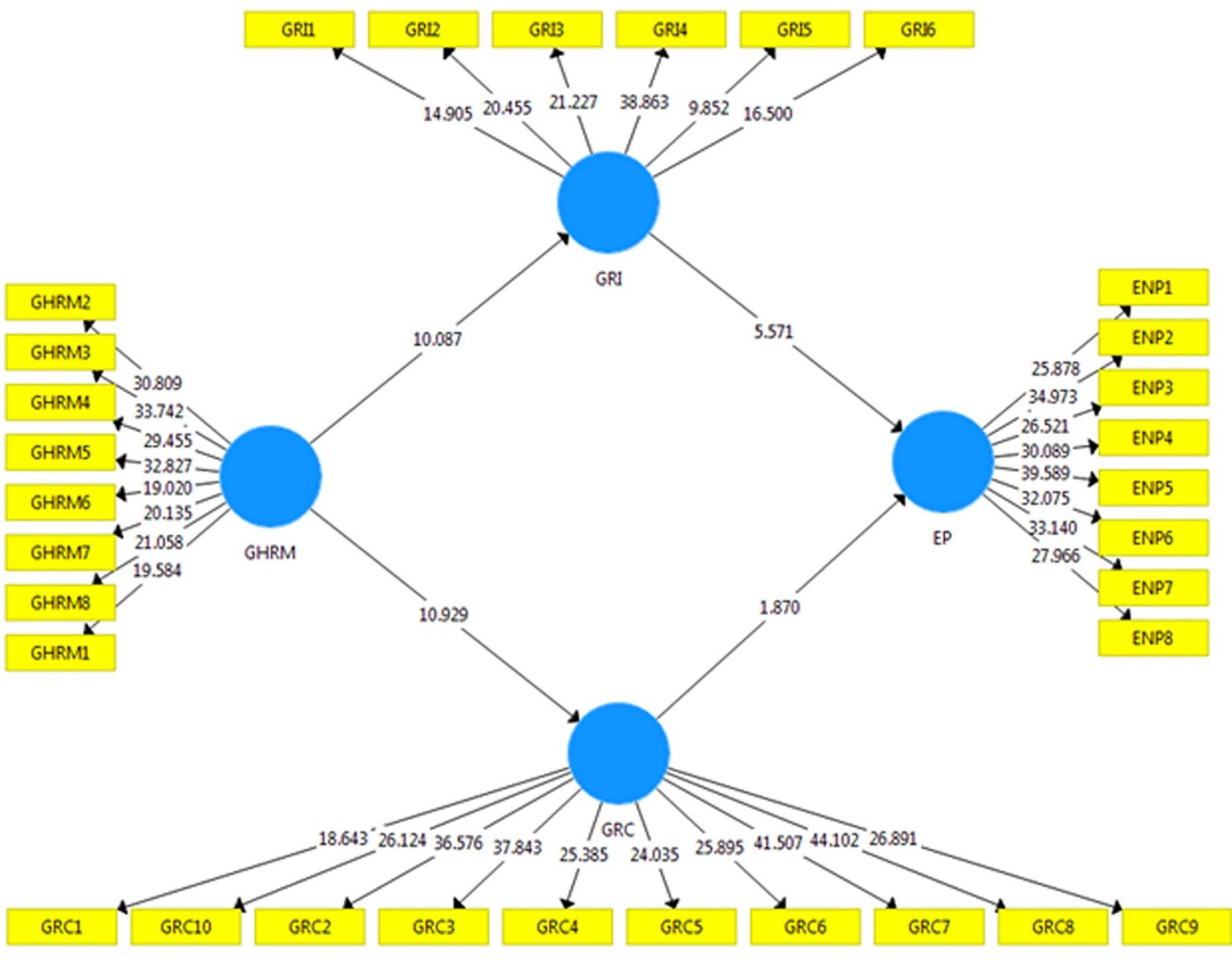

**Fig 3. Structural model.**

## Structural model

As a first step, a direct relationship between green human resource management (GHRM), green innovation (GI), and green culture (GC) was investigated in order to calculate the direct influence of these variables on environmental performance (EP). Furthermore, the indirect relationship was also investigated in Fig 3 and Table 5 shows beta values and the t-value to determine whether the hypotheses are supported or not.

The hypothesis testing portion of this regression test is addressed the direct and indirect relationship of variables. Table 5 shows the beta-value, t-value, and p-value used to determine

**Table 2. Constructs' reliability and convergent validity.**

| Variables | Cronbach's α | rho_A | C.R | AVE |
|---|---|---|---|---|
| EP | 0.921 | 0.924 | 0.935 | 0.642 |
| GHRM | 0.915 | 0.915 | 0.931 | 0.631 |
| GRC | 0.926 | 0.927 | 0.938 | 0.602 |
| GRI | 0.825 | 0.837 | 0.873 | 0.535 |

Note(s): EP environmental performance; GHRM green human resource management; GRC green culture; GRI green innovation.

**Table 3. Fornell–Larcker for discriminant validity.**

| Variable | EP | GHRM | GRC | GRI |
|---|---|---|---|---|
| EP | 0.801 | | | |
| GHRM | 0.379 | 0.794 | | |
| GRC | 0.363 | 0.538 | 0.776 | |
| GRI | 0.462 | 0.493 | 0.581 | 0.732 |

Note(s): EP environmental performance; GHRM green human resiurce management; GRC green culture; GRI green innovation.

**Table 4. Heterotrait–monotrait ratio (HTMT) for first-order.**

| Variable | EP | GHRM | GRC | GRI |
|---|---|---|---|---|
| EP | | | | |
| GHRM | 0.399 | | | |
| GRC | 0.381 | 0.575 | | |
| GRI | 0.485 | 0.557 | 0.687 | |

Note(s): EP environmental performance; GHRM green human resiurce management; GRC green culture; GRI green innovation.

**Table 5. Direct and indirect effects of hypothesis.**

| Hypotheses | Paths | B-value | S-Deviation | t-values | p-values | Remarks |
|---|---|---|---|---|---|---|
| 1 | GHRM -> EP | 0.264 | 0.040 | 6.561 | 0.000 | Supported |
| 2 | GHRM -> GRI | 0.493 | 0.049 | 10.087 | 0.000 | Supported |
| 3 | GHRM -> GRC | 0.538 | 0.049 | 10.929 | 0.000 | Supported |
| 4 | GRI -> EP | 0.379 | 0.068 | 5.571 | 0.000 | Supported |
| 5 | GRC -> EP | 0.143 | 0.066 | 1.970 | 0.042 | Supported |
| 6 | GHRM -> GRI -> EP | 0.157 | 0.038 | 4.883 | 0.000 | Supported |
| 7 | GHRM -> GRC -> EP | 0.277 | 0.043 | 1.972 | 0.047 | Supported |

Note(s): EP environmental performance; GHRM green human resiurce management; GRC green culture; GRI green innovation.

whether hypotheses are accepted or rejected. Furthermore, this part has a structural model that can be used to evaluate research hypotheses. The direct and indirect hypotheses results are shown in Table 5 of this portion. The structural model is shown in Fig 3. GHRM has a positive relationship with environmental performance (p-value 0.000; t-value 6.561), and the evidence has validated H1. While H2, GHRM and green innovation (p-value 0.00; t-value 10.087) and H3, GHRM and green culture value (p-value 0.000; t-value 10.929) were both supported (p-value 0.000; t-value 5.571) H4 is supported, H5, the relationship between green culture and environmental performance value (p-value 0.042; t-value 1.970) is supported. Furthermore, green innovation is shown to be a significant mediator of the relationship between green human resource management (GHRM) and environmental performance (p-value 0.000; t-value 4.883). It is found to support hypothesis H6. The variance accounted for (VAF) was employed to establish the mediating effects of green innovation on GHRM and environmental performance. In the case of VAF values less than 20 percent, between 20 and 80 percent, and higher than 80 percent, the value of no mediation, partial mediation, and full mediation, in

**Table 6. Predictive relevance ($Q^2$) and effect size ($f^2$).**

| $F^2$ | EP | GRC | GRI | $Q^2$ | $R^2$ |
|---|---|---|---|---|---|
| EP | | | | 0.139 | 0.227 |
| GHRM | | 0.407 | 0.322 | | |
| GRC | 0.018 | | | 0.169 | 0.289 |
| GRI | 0.123 | | | 0.132 | 0.243 |

Note(s): EP environmental performance; GHRM green human resource management; GRC green culture; GRI green innovation.

that order, has been determined. The mediating effect is the same as if it were to fall under the partial mediation criterion. As a result, H6 is only partially mediated. Furthermore, H7 green culture mediation has no statistically significant effect on GHRM and environmental performance, whereas their values are (p-value 0.047; t-value 1.972) and are not supported by H7 green culture mediation.

In this case, the value of $f^2$ indicates whether an exogenous variable affects an endogenous variable [131]. Cohen suggests that the impact of $f^2$ be classified into three categories: minor effect ($f^2 = 0.02$), medium effect ($f^2 = 0.15$), and large effect ($f^2 = 0.35$) (1988). Table 6 demonstrates that the GHRM value has a larger impact on green culture ($f^2 = 0.407$) and a medium impact on green innovation ($f^2 = 0.322$). Green culture ($f^2 = 0.018$) and green innovation ($f^2 = 0.123$) have a smaller effect on environmental performance than the other variables. Furthermore, as stated by [132], the value of $Q^2$ must be higher than zero in order to be valid. $Q^2$ meets all of the criteria mentioned above, according to the current study, as shown in Table 6. $Q^2$'s environmental performance is 0.139, while its green innovation is 0.132, as shown in the present research. Our study also reveals that $Q^2$ of green culture is 0.169 larger than zero and has a strong predictive power at the variable level, which is a significant finding. $R^2$ was evaluated in the research in order to calculate the model's explanatory power for the variables. According to [133], when it comes to the model's explanatory power, an $R^2$ value of 0.10 is considered to be the bare minimum desired in social science research contexts. Table 6 shows that all $R^2$ values are significantly more than the minimal requirement, confirming our findings.

## Discussion and conclusion

According to our knowledge, for the first time, this study emphasizes the relationship between GHRM and environmental performance by addressing green innovation and green culture as a new approach to enhancing environmental performance in the perspective of the manufacturing sector in Malaysia. To put it another way, our study adds to the expansion of existing study on HRM and environmental management by examining the impact of GHRM practices on environmental performance in Malaysian manufacturing contexts with the mediationg role of green innovation and green culture. As per our conclusions, organizations' GHRM practices positively impact their environmental performance. As a result, environmental performance can be enhanced by changing workers' attitudes and behaviours. Results are similar to earlier research [5, 14, 16], which show that GHRM practices can help employees build greener thinking and encourage him to involve in eco-friendly behaviour [8, 16]. Thus, our findings show that an organization's pro-environmental attitude can lead to better environmental performance.

Our findings show that the GHRM practises of the manufacturing firms in Malaysia are positively related to their environmental performance, which is in line with our predictions.

Several factors contribute to this, including that improving environmental performance can be accomplished by changing employees' attitudes and behaviours inside an organization. GHRM practises, according to the outcomes of other investigations [9, 60, 134], suggest that GHRM support the development of greener minds and the motivation of employees to engage in environmentally friendly activities. Our findings, therefore, demonstrate that pro-environmental performance behaviour within the organization can increase to enhanced environmental performance results in the long term.

In a similar vein, our research demonstrates that GHRM practises tremendously boost green innovation, increasing environmental performance. GHRM practises significantly improve Green innovation and environmental performance. Or to put it another way, green innovation playing a key function as a in mediator in their relationship between green HRM practices and environmental performance. Prior research by [18, 19, 61] found that green HRM practices are extremely important to boosting green employee innovation and that innovation for the environment can contribute to employees' additional-role behaviour and increase their performance in terms of environmental improvement [135].

Aside from that, our study reveals that a green organizational culture in organizations is essential in understanding the correlation between GHRM and an organization's EP. In particular, we have noticed that improving a firm's environmental performance necessitates the adoption of pro-environmental GHRM practices [136]. Green culture is a positive mediating factor in the association with GHRM and EP. According to the findings, pro-environmental GHRM activities help to promote the development of a green culture. As per our conclusions, the green culture supports employees to proactively decline waste, use minimum resources, and adapt recycling programmes, ultimately improving the organisation's environmental performance. This conclusion is compatible with the findings of [137, 138], which demonstrated that green culture plays a vital role and has a beneficial impact on environmental performance, respectively. We believe that this is the unique research investigation to explore green innovation and green culture as mediating factors in the interrelation with GHRM and EP in Malaysian manufacturing.

## Implications of these results

Our research findings have main implications for both managers and academics who can put on green culture to the next generation of environmentally conscious managers. This study has implications for management in that it can assist managers in persuading staff to undertake pro-environmental measures in their everyday jobs. Our results imply that HR managers can employ pro-environmental recruiting and training to help build a green culture in their organizations. Contracting environmentally sensitive personnel and then putting in place a regular, efficient training and monitoring machanism help raise environmental understanding within the organization's different roles. These actions help to guarantee that environmental cognizance is ingrained in the behaviours and habits of the organization's workforce. These behaviours develop routines over time, which can help to build a pro-environmental culture inside the firm [8]. The result is that employees' endeavours to commence environmentally responsible activities to improve their organization's environmental performance are bolstered as a result of this culture. To this end, we recommend that managers consider GHRM activities while driving environmental performance gains and the crucial part that culture shows in the long-term growth of their organizations.

Due to a lack of empirical data and must-know criteria for new staff can use to learn about green management, organizations can face a challenge in passing on their green organizational culture and green innovation to the new recruits of liable managers. This is because a large

number of the studies on GHRM promotes green organizational culture and innovation as key topics without providing a wide variety of statistical verified data to support these conclusions. Because we propose and evaluate a distinctive framwork based on statistical data gathered from one of the significant economies in today's world wide: Malaysia, our article can be valuable in developing the new employee's knowledge of green organizational management inside organizations in this environment. We provide more information on the relationship between GHRM, GI, and GC by revealing environmental performance. It will be conceivable to dispute whether or not these essential factors act as a mediating factor in the relationship between GHRM and environment performance. This in-depth debate can benefit academicians who are teaching and studying a particular topic. In this way, academicians in charge of educating upcoming generations of more responsible higher managers about green organizational culture and GI will consider in this research a valuable source of data to include in the component guideline on green management, which is affiliated with the learning of next generations of more concerned managers (Muisyo et al., 2021).

## Limitations and future research

To be sure, our research has certain limitations, which we acknowledge. Despite the large number (290) of participants in this study, the sample size is still limited when compared to the overall population of Malaysia's manufacturing sector. As a consequence of the limited sample, the generalizability of the findings may be restricted in certain ways. As a result, we acknowledge that our study assessed green culture rather than concentrating on the components of organizational culture, as suggested by [11, 139]. According to [140], future research would necessary to deliberate the overall pro-environmental attitudes and principles and activities to provide a broad overview of green organizational culture in the workplace. Managerial perceptions of environmental initiatives [141]; the modernization of green innovation [142]; and the consistency of top management beliefs concerning environment protection are some values and beliefs that should be considered in future studies [143]. It is also possible to investigate in further depth the role played by green culture as well as green innovation in encouraging voluntary green workplace behaviour [144]. Organizational culture and innovation in sustainable development research have already been identified as a major challenge in the literature [145, 146], and we believe that additional research in this important area is highly required.

## Author Contributions

**Conceptualization:** Shengxu Shi.

**Data curation:** Liuyue Fang, Shengxu Shi.

**Formal analysis:** Liuyue Fang.

**Funding acquisition:** Liuyue Fang.

**Investigation:** Shengxu Shi, Jingzu Gao, Xiayun Li.

**Methodology:** Liuyue Fang, Shengxu Shi, Jingzu Gao, Xiayun Li.

**Resources:** Jingzu Gao.

**Supervision:** Xiayun Li.

**Writing – original draft:** Liuyue Fang.

**Writing – review & editing:** Liuyue Fang.

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
