## [Decision Letter · Decision Letter 0]

27 Apr 2022

PONE-D-22-00680The Mediating Role of Green Innovation and Green Culture in the Relationship between Green Human Resource Management and Environmental PerformancePLOS ONE

Dear Dr. Fang,

Thank you for submitting your manuscript to PLOS ONE. After careful consideration, we feel that it has merit but does not fully meet PLOS ONE’s publication criteria as it currently stands. Therefore, we invite you to submit a revised version of the manuscript that addresses the points raised during the review process.

We look forward to receiving your revised manuscript.

Kind regards,

Alessandro Margherita

Academic Editor

PLOS ONE

Journal Requirements:

3.  a. Thank you for including your ethics statement:  "FULL ETHICS STATEMENT HERE".  

For studies reporting research involving human participants, PLOS ONE requires authors to confirm that this specific study was reviewed and approved by an institutional review board (ethics committee) before the study began. Please provide the specific name of the ethics committee/IRB that approved your study, or explain why you did not seek approval in this case.

       b. Please provide additional details regarding participant consent. In the ethics statement in the Methods and online submission information, please ensure that you have specified (1) whether consent was informed and (2) what type you obtained (for instance, written or verbal, and if verbal, how it was documented and witnessed). If your study included minors, state whether you obtained consent from parents or guardians. If the need for consent was waived by the ethics committee, please include this information.

6. Thank you for stating the following in your Competing Interests section: 

“No competing interest”

Reviewers' comments:

Reviewer's Responses to Questions

**Comments to the Author**

1. Is the manuscript technically sound, and do the data support the conclusions?

Reviewer #1: Yes

Reviewer #2: Yes

2. Has the statistical analysis been performed appropriately and rigorously? 

Reviewer #1: Yes

Reviewer #2: Yes

3. Have the authors made all data underlying the findings in their manuscript fully available?

Reviewer #1: No

Reviewer #2: No

4. Is the manuscript presented in an intelligible fashion and written in standard English?

Reviewer #1: Yes

Reviewer #2: Yes

5. Review Comments to the Author

Reviewer #1: Expand literature background (and deriving assumptions for the study) by including a more in-depth investigation of related topcis such as CSR (Corporate Social Responsibility) and circular economy/business models.

Illustrate better the nature (more than just the composition) of the sample and explain better how results can be generalized or not (you mention in the limitations section but need to be more specific on why the paper is anyhow a contribution to academic/practitioner discussion).

Enhance discussion with a more robust and in-depth investigation of where/how the study advances current knowledge, also at the light of the research gap identified in the introduction.

Have the paper proofread by a mothertongue.

Reviewer #2: More explanation of methodology analysis needed. For example, what model it was used, which software, what types of test were conducted and why?.. I believe the limitation section need to include the low level of AVE and why you do think this low level appears and how future studies could try resolve that. What ethical procedures were followed?

6. PLOS authors have the option to publish the peer review history of their article (what does this mean?). If published, this will include your full peer review and any attached files.

Reviewer #1: No

Reviewer #2: **Yes: **Ahmed Asfahani

---

## [Author Response · Author response to Decision Letter 0]

18 Aug 2022

Date: Apr 27 2022 03:13AM

To: "Liuyue Fang" fly0612@sina.com

From: "PLOS ONE" plosone@plos.org

Subject: PLOS ONE Decision: Revision required [PONE-D-22-00680]

PONE-D-22-00680

The Mediating Role of Green Innovation and Green Culture in the Relationship between Green Human Resource Management and Environmental Performance

PLOS ONE

Dear Dr. Fang,

Thank you for submitting your manuscript to PLOS ONE. After careful consideration, we feel that it has merit but does not fully meet PLOS ONE’s publication criteria as it currently stands. Therefore, we invite you to submit a revised version of the manuscript that addresses the points raised during the review process.

---

## [Editor Report · Decision Letter 1]

6 Sep 2022

The Mediating Role of Green Innovation and Green Culture in the Relationship between Green Human Resource Management and Environmental Performance

PONE-D-22-00680R1

Dear Dr. Fang,

We’re pleased to inform you that your manuscript has been judged scientifically suitable for publication and will be formally accepted for publication once it meets all outstanding technical requirements.

Kind regards,

Alessandro Margherita

Academic Editor

PLOS ONE
---

## [Editor Report · Acceptance letter]

19 Sep 2022

PONE-D-22-00680R1 

The mediating role of green innovation and green culture in the relationship between green human resource management and environmental performance 

Dear Dr. Fang:

I'm pleased to inform you that your manuscript has been deemed suitable for publication in PLOS ONE. Congratulations! Your manuscript is now with our production department. 

Kind regards, 

on behalf of

Dr. Alessandro Margherita 

Academic Editor

PLOS ONE